# Bayesian spatio-temporal modeling of malaria risk in Rwanda

**Muhammed Semakula**[1,3]*, **François Niragire**[2], **Christel Faes**[1]

**1** I-BioStat, Hasselt University, Hasselt, Belgium, **2** Department of Applied Statistics, College of Business and Economics, University of Rwanda, Kigali, Rwanda, **3** College of Business and Economics, Center of excellence in Data science, Bio-statistics, University of Rwanda, Kigali, Rwanda

* semakulam@gmail.com

## Abstract

Every year, 435,000 people worldwide die from Malaria, mainly in Africa and Asia. However, malaria is a curable and preventable disease. Most countries are developing malaria elimination plans to meet sustainable development goal three, target 3.3, which includes ending the epidemic of malaria by 2030. Rwanda, through the malaria strategic plan 2012-2018 set a target to reduce malaria incidence by 42% from 2012 to 2018. Assessing the health policy and taking a decision using the incidence rate approach is becoming more challenging. We are proposing suitable statistical methods that handle spatial structure and uncertainty on the relative risk that is relevant to National Malaria Control Program. We used a spatio-temporal model to estimate the excess probability for decision making at a small area on evaluating reduction of incidence. SIR and BYM models were developed using routine data from Health facilities for the period from 2012 to 2018 in Rwanda. The fitted model was used to generate relative risk (RR) estimates comparing the risk with the malaria risk in 2012, and to assess the probability of attaining the set target goal per area. The results showed an overall increase in malaria in 2013 to 2018 as compared to 2012. Ofall sectors in Rwanda, 47.36% failed to meet targeted reduction in incidence from 2012 to 2018. Our approach of using excess probability method to evaluate attainment of target or identifying threshold is a relevant statistical method, which will enable the Rwandan Government to sustain malaria control and monitor the effectiveness of targeted interventions.

## 1 Introduction

Malaria remains a public health threat in developing countries, even though it is a preventable and curable disease. Every two minutes, the life of a child under age five is lost due to the disease [1]. There are a total of 435,000 deaths per year because of malaria, mainly in Africa and Asia [2]. Though some countries have successfully eliminated malaria, those with a high burden of disease have recorded an increase in malaria cases for the last decade. Sub-Saharan Africa and India contributed eight percent to the global burden [2].

The World Health Organization (WHO) Global Technical Strategy for malaria (GTS) aims to eliminate malaria worldwide by 2030. WHO classified countries and communities based on

**Data Availability Statement:** All relevant data are within the paper and its Supporting Information files.

**Funding:** The authors received no specific funding for this work.

**Competing interests:** The authors have declared that no competing interests exist.

progress towards elimination of malaria (Control or Elimination). Malaria elimination is defined as the interruption of local transmission by reducing the rate of malaria cases to zero for a specific malaria parasite in a defined geographic area over particular time period. Malaria control is defined as a reduction of disease incidence, prevalence, morbidity or mortality to a locally acceptable level as a result of deliberate efforts. Most countries have placed malaria elimination by 2020 on their health agenda, though fewer than 30 countries worldwide were certified malaria-free by WHO in the last 60 years [3, 4].

The Malaria elimination feasibility studies proved that it can be eliminated. Elimination of malaira requires a strong health system that where communities have to access quality services, strong health information systems for tracking progress, effective surveillance, and system for public health response [3].

The Malaria Strategic Plan (MSP) 2012-2018 contained ambitious goals aimed at eliminating malaria death and reducing malaria morbidity by 2018, with a testing yield less than 5% test positivity rate by 2018 [5]. Contrary to expectation, the number of malaria cases increased in Rwanda, with 10 times more cases in 2017 as compared to 2011. The increase in malaria cases is often associated with the direct and indirect influence of climate change [6]. The 2016 Mid-Term Review Report (MTR) of MSP concluded that it is unlikely that Rwanda will meet the pre-elimination objectives and recommended not to review applicability and implementation of pre-elimination in line with WHO Guidelines. The MTR acknowledged the performance level of health management information system (HMIS) [5, 7].

The Rwanda health sector strategic plan (HSSP III) presented five key strategies for the pre-elimination phases and five indicators to be tracked that including (1) reducing malaria prevalence among women and children under-five, (2) the reduction of malaria incidence from 26 per 1000 in 2011 to 20 per 1000 in 2015 and to 15 per 1000 in 2018 (with a positivity rate less than 5%), (3) increasing the number children under five sleeping in Long-Lasting Insecticidal Nets(LLIN) to 82% in 2018 from 15% in 2011, (4)reducing malaria proportional morbidity from 4 to 3 in 2018; (5) and increasing the percentage of households with at least 1 LLIN installed from 82% to above 85% in 2018 [8].

The elimination of malaria requires a strong surveillance system to detect malaria infections early and enable a rapid and effective response. The World Health Organization and Global Fund promote the use of a health information system. Most developing countries adopted District Health Information Software (DHIS) [9]. The DHIS is a free and open source platform for the management of routine health information with a primary focus on producing health statistics [10]. Rwanda's health system uses DHIS for data recording, reporting, and analysis. The statistical analyses offered by DHIS include basic descriptive statistics and data visualizations. For the epidemiological surveillance of malaria, HMIS enables aggregation of data in one platform from all health facilities in Rwanda. Those data are used for further statistical analysis to inform evidence-based strategies to control malaria. The Rwanda Malaria control program uses WHO recommended operational methods to detect the epidemic threshold. The method is to compare the constant case count with mean ± 2 SD (standard deviation) or median + the upper third quintile of the previous year's series data [11]. The incidence maps used for decision making rely on a fixed cut off to determine a high or low incidence rate. However, none of those estimation methods take into consideration the spatial uncertainty or account for the population at risk. Nevertheless, those methods are sensitive to outliers and unlikely to detect malaria patterns in low transmission areas [12]. These approaches can help to visualize the overall dispersion around prevalence or incidence estimates but do not provide any information linked to the uncertainty of exceeding probability or incidence threshold [13].

Currently, there is an increase in the use of model-based approaches with data from surveys as suggested by authors of the feasibility of the malaria elimination phase [14]. The surveys are often inadequately powered to detect very low levels of heterogeneous transmission and those surveys are performed periodically, most often every five years. In contrast, routinely collected clinical data are timely and local. Few studies have combined model-based approaches, routinely collected clinical data, and population census data to inform national malaria elimination efforts.

A model-based approach of studying geospatial malaria trends is useful in identifying risk factors in the general population and informing evidence based-decisions. The statistical models allow the inclusion of a variety of features that capture the variation of disease risk [15]. In this paper, spatio-temporal methods will be used to investigate the geographical variation of malaria risk. We use routinely collected malaria data from health facilities in each sector of Rwanda to illustrate a formal assessment of pre-specified target goals, which can be used to evaluate reduction of incidence and progress towards targets. Understanding geographic disparities at a broad level is useful to a certain extent, but is unlikely to accurately reflect the heterogeneity in outcomes at the local level [16]. Malaria elimination efforts can benefit greatly by quantifying the variation across population groups and small geographical areas. An understanding of the geographic patterns of malaria can inform decision making by the government, and non-governmental organizations for policy development, targeted interventions and the adequate allocation resources at the area with the most acute need.

## 2 Materials and methods

### 2.1 Data source

We used data on malaria cases from the Rwanda health information system (HMIS) for the period from January 1, 2012 through December 31,2018. Over 95% of malaria cases reported through HMIS in Rwanda are laboratory confirmed [17]. Data on the number of malaria cases are available at the level of the health centre and are disaggregated by sex and age. Rwanda's health system is organized into hierarchy of five levels: (1) referral hospitals provide the highest levels of specialty care, followed by (2) district hospitals and (3) health centers at the sector level. Below health centres are community-based health services including (4) health posts and (5) community health workers. Rwanda has 416 administration sectors and each has at least one health centre. For this analysis, we analyzed malaria cases at the sector level. For the children under five, which constituted 12% of cases, data is not disaggregated by sex. Therefore, these cases were excluded from the analysis.

Population data for 2012 were taken directly from the 2012 census. For population estimates in the period from 2013 to 2018. We used projections based on the 2012 census. Population data were downloaded from the following link www.statistics.gov.rw/datasource/42.

### 2.2 SIR

We adapted the traditional approach of calculating the Standardized Incidence Ratio (SIR) in each area $i$ ($i = 1, \ldots, n$) and year $t$ ($t = 2012, 2013, \ldots, 2018$), correcting for the age, and gender- demographic structure in an area. We will use the SIR as a tool to investigate the change in malaria risk at time $t$ as compared to a certain reference year, in our case, the year 2012. As result, we define the SIR as the ratio of the number of observed cases $y_{it}$ to the number of expected cases $E_{it}$ in the $i^{th}$ area at time $t$:

$$SIR_{it} = \frac{y_{it}}{E_{it}},$$

(1)

with the expected number of cases calculated as

$$E_{it} \quad = \quad \sum_{j=1}^{J} N_{ijt} r_j \qquad (2)$$

the $r_j$ is the reference rates in age and gender-group $j$ and $N_{ijt}$ is the population in the area $i$, age-gender group $j$ and time $t$:

$$r_j = \frac{y_j^{2012}}{N_j^{2012}} \qquad (3)$$

where $y_j^{2012}$ are the cases observed in age/gender group $j$ in Rwanda in 2012, and $N_j^{2012}$ is the census population for 2012 in Rwanda in the corresponding age/gender group.

To evaluate progress towards the targeted reduction of malaria incidence in the Malaria strategic plan 2012-2018, the reference rate is based on the malaria incidence in year 2012. This will enable comparison of malaria rates with subsequent years. The expected counts therefore represent the total number of malaria cases that one would expect if the population in area $i$ contracted the disease at the same rate as in 2012.

## 2.3 Model specification

As SIR uses information only from within an area, it might produce uncertain estimates for small areas. Classical methods do not take into account the spatial dependence among the areas. Therefore, we used Bayesian disease mapping approaches that take into account the spatial dependence among neighboring areas.

A Bayesian disease mapping model consists of three components: the data model (i.e. the distribution of the data given the parameters), the process model (i.e. a description of underlying spatial trend) and the parameter model (i.e. the prior distribution of the parameters to be estimated) [18]. The data model is given by

$$Y_{it} \quad \sim \quad \text{Poisson}(E_{it}\theta_{it}), \qquad (4)$$

where a Poisson distribution is appropriate since disease data are counts (number of cases and are non-negative). It is assumed that the mean is a product of the expected count $E_{it}$ and the relative risk $\theta_{it}$.

The process model describes the underlying structure of the relative risks. We used the spatio-temporal extension of the spatial Besag-York-Mollie (BYM) model, which is the CAR convolution model with two random effects, one spatially-structured area-specific random effect and one unstructured area-specific random effect [19, 20]

$$log(\theta_i) = \alpha + u_i + v_i + \gamma_t + \psi_t + \delta_{it} \qquad (5)$$

where, $u_i$ is the spatially-structured area-specific random effect which allows for smoothing amongst adjacent areas, namely [19]

$$u_i | u_j \sim N\left(\bar{\mu}_{\delta_i}, \frac{\sigma_u^2}{n_{\delta_i}}\right)$$

with $\delta_i$ and $n_{\delta_i}$ respectively, the set of neighbours and number of neighbours for a specific area $i$. The unstructured component $v_i$ is modeled using as a Gaussian process

$$v_i \sim N(0, \sigma_v^2),$$

and allows for extra heterogeneity in the counts due to unobserved (and spatially unstructured) risk factors. The $\gamma_t$ term represents the temporally structured effect, modeled dynamically using random walk of order 2 (RW of order 2) and defined as

$$\gamma_t | \gamma_{t-1}, \gamma_{t-2} \sim N(2\gamma_{t-1} + \gamma_{t-2}, \sigma^2).$$

The term $\psi_t$ is specified by means of Gaussian exchangeable prior, defined as $\psi_t \sim N(0, \frac{1}{\tau_\psi})$.

In order to allow for interaction between space and time, which explain differences in the time trend of malaria risk for different areas, the parameter $\delta_{it}$ follow a Gaussian Distribution with a precision matrix given by $\tau_\delta \mathbf{R}_\delta$, where $\tau_\delta$ is unknown scalar, while $\mathbf{R}_\delta$ is the structure matrix, identifying the type of temporal and/ or spatial dependence between the elements of $\delta$. $\mathbf{R}_\delta$ can be factorized as the Kronecker product of the structure matrix of corresponding main effects which interact. There are four ways to define the structure matrix as presented in literature [21] and reported in Table 1. We fitted models that consider three different types of interactions.

The best model was chosen basing on deviance information criterion (DIC) [18], sensitivity analysis and condition predictive ordinate (CPO) [22]. The DIC is a popular choice for model selection, although it has been demonstrated that DIC might be problematic in practice [23]. It is based on the productive accuracy of the estimated model, connecting for the number of parameters to be estimated after incorporating the prio information. The CPO is defined as $CPO_i = \pi(y_i^{obs} | y - i)$; $y - i$ denotes the observations $y$ with the $i$-th component omitted. Therefore, it is leave one out cross-validation scare. it expresses the posterior probability of observing value $yi$, when the model is fitted to all data except $yi$. Based on the CPO-values a logarthmic score is defined as $-\sum logCPO_I$. Smaller values of this scare indicates a better predictive quality of the model. The CPO are computed after description of models in R-INLA routenely via importance sampling without rerunning the model [24].

The type I interaction corresponds to a combination of an independent spatial and temporal random effect. As the spatial effect, temporal effect are assumed to be independent, an interaction of those have the correlation matrix $\mathbf{R}_\delta = \mathbf{R}_v \otimes \mathbf{R}_\psi = \mathbf{I} \otimes \mathbf{I} = \mathbf{I}$,. Thus, we assume no spatial and /or temporal structure on the interaction either and therefore $\delta_{it} \sim Normal(0, \frac{1}{\tau_\delta})$.

The Type II interaction combines a structured temporal effect with an unstructured spatial effect. The structure matrix therefore is defined as $\mathbf{R}_\delta = \mathbf{R}_v \otimes \mathbf{R}_\gamma$, where $\mathbf{R}_v = \mathbf{I}$ and $\mathbf{R}_\delta$ is the neighborhood structure specified for instance through a first or second order random walk. This leads to an interaction term which is temporally correlated whith each spatial unit, while the time trends in the defferent areas are independent. The Type III interaction combines an unstructured temporal effect with a spatially structured effect. The structure matrix is defined as $\mathbf{R}_\delta = \mathbf{R}_\psi \otimes \mathbf{R}_u$, where $\mathbf{R}_\psi = \mathbf{I}$ and $\mathbf{R}_u$ is a neighboring defined through the CAR specification. This leads to the assumption that the parameters $t' \neq t$ at time point t $\{\delta_t, \ldots, \delta_{nt}\}$, have a spatial structure independent from the other time points.

We assigned a gamma distribution with shape equal 0.5 and rate equal to 0.00149 following the approach of Fong et al.(2010) [25] and it was not sensitive to arbitrary choices after sensitivity analysis. For the remaining parameters, we assigned prior distributions to scaled

Table 1. Interaction types: Parameter interacting and rank of $\mathbf{R}_\delta$.

| Type of interaction | structure matrix | Rank |
|---|---|---|
| Type I interaction | $\mathbf{R}_\delta = \mathbf{R}_v \otimes \mathbf{R}_\psi = I \otimes \mathbf{I} = \mathbf{I}$ | $nT$ |
| Type II interaction | $\mathbf{R}_\delta = \mathbf{R}_v \otimes \mathbf{R}_\gamma$ | $n(T-2)$ for RW2 |
| Type III interaction | $\mathbf{R}_\delta = \mathbf{R}_\psi \otimes \mathbf{R}_u$ | $(n-1)T$ |

precision matrix parameters based on their marginal standard deviations on its diagonal following methods proposed by Sorby and Rue (2013) [26].

In order to investigate whether or not a reduction of malaria was observed compared to the overall incidence rate in 2012, we make use of excess probability. The probability that the malaria risk has decreased by c% is calculated as the posterior probability $P(\theta_{it} < (100 - c)\%)$. If $|P|$ is large, the set goal is likely reached in that area, while if $|P|$ small, it is very likely that is has not been reached.

## 2.4 Estimation methods

We used Integrated Nested Laplace approximation (INLA) for estimation. The INLA is a deterministic algorithm for Bayesian inference and is designed for latent Gaussian models and spatial models. Bayesian estimation using the INLA methodology takes much less time as compared to estimation using Markov Chain Monte Carlo Methods (MCMC) [27].

We performed a sensitivity analysis on a variety of model formulations for the latent level due to the inherent issues that come with each formulation. It is well known from literature that in the BYM model, the spatially structured component cannot be seen independently from the unstructured component. BYM2 model is an alternative to improve parameter control by allowing the parameter to be seen independently from each other [26]. We fitted both models (BYM and BYM2) using the same priors. Results from both models were similar.

## 3 Results

The results are presented into two parts. The first part provides summary descriptive statistics of malaria cases and estimates from the fitted spatio-temporal model. The second part presents an evaluation of Rwanda's malaria policy on the reduction of incidence using the excess probability approach. We introduced formal friendly interpretation and classification based on the excess probability approach for decision making during the malaria pre-elimination phase.

### 3.1 Malaria cases in Rwanda 2012-2018

Rwanda experienced an increase in malaria cases from 2012 to 2016, with 398,287 cases during 2012 to 2,956,337 cases in 2016. However, during 2017 and 2018, the total number of cases decreased to 1,978,450 and 1,725,522 respectively. Fig 1 shows the overall trend as well as the trend by age groups and sex from 2012 to 2018. The highest number of cases were reported in all age and sex groups in 2015 and 2016.

### 3.2 Malaria relative risk in Rwanda 2012-2018: BYM

We have fitted spatio-temporal models for the period from 2012-2018, taking into account both structured and unstructured random effects (BYM and BYM2 models) as it provides a compromise between spatial correlation and extra heterogeneity over time. Since the results of those models are similar, we present the BYM model fitted with type II interaction based on Deviance Information Criterion (DIC) and Watanabe Akaike Information Criterion (WAIC) 2. The DIC is a tool for model selection in Bayesian context [28]; we computed a Bayesian measure of complexity (pD) and Bayesian deviance (D), DICc adjusted and WAIC that is also used for Bayesian model selection [29]. Both DIC and WAIC appraoch suggested mod.intII as the best model compared to others models fitted as Table 2 shows. In addition, we used predictive ordinate (CPO) for cross-validation of model.

Those models provide the estimates at the smallest available geographical scale, that might be an added value to drive oriented and targeted interventions to control malaria in Rwanda.

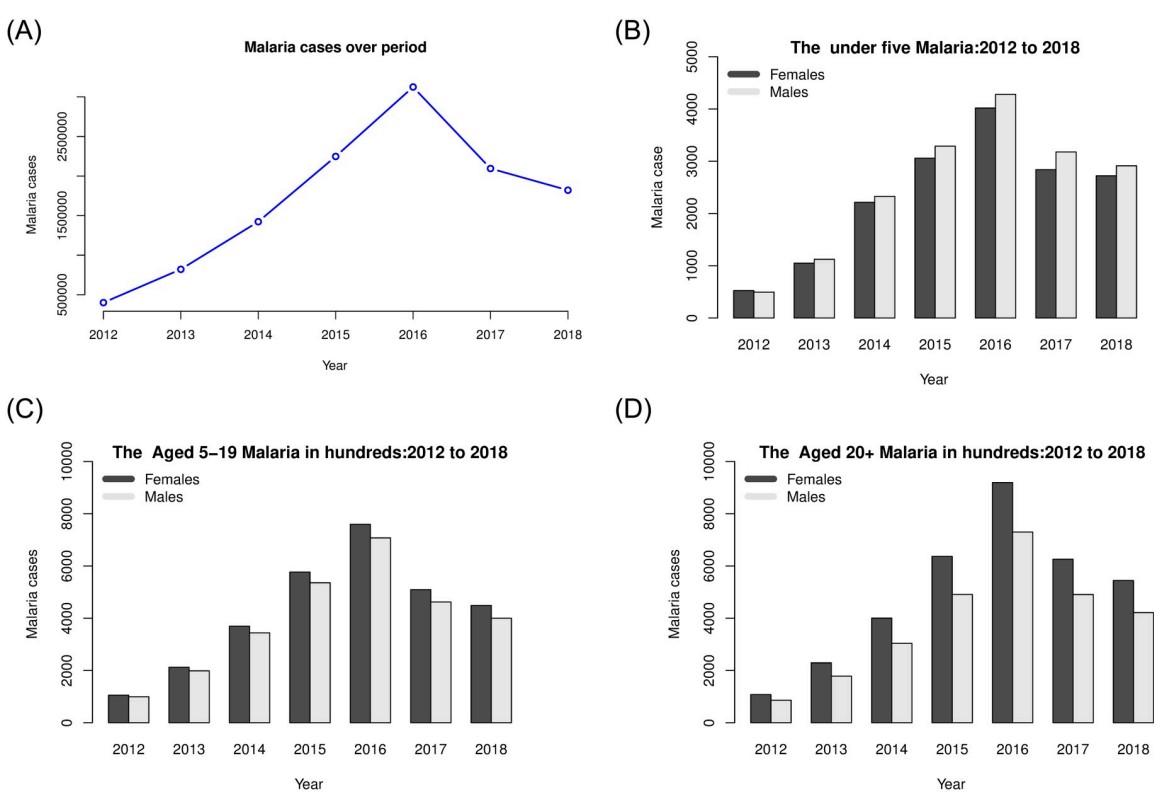

**Fig 1. Malaria cases over time by sex.**

Estimates of variances due to random effects are presented in Table 3, the contribution of variance can be summarized, as follows: approximately 50% is explained by a spatial component, and 50% by an unstructured component. This is also visible in Fig 2, which presents estimated relative risks for each year, compared with the overall incidence rate year in 2012.

Fig 3 shows an increasing trend effect for malaria relative risk in Rwanda with 95% Credible Interval over years.

In general, the spatio-temporal contribution to geographic variability is important, as there is a tendency to see low relative risks in the North-West of Rwanda, and high relative risk in the East and in the South of Rwanda. We also observe a large amount of heterogeneity amongst areas, as some of the areas with high relative risk for malaria are surrounded by areas with low risk (and vice versa). Table 4 shows the number of sectors with RR's within specific intervals.

In 2012, 73.8% (307) of all sectors (416) had a RR < 1,(a lower than average disease rate), while 18.03% (75) of the sectors had a RR above one but below 4, and 5.53% (23) had a RR above 4 but below 10. In 11 sectors, the RR was above ten, including four sectors with a RR

**Table 2. Comparison of models basing on DIC and WIAC.**

| Model | D | pD | DIC | DICc | WAIC |
|---|---|---|---|---|---|
| model.ST1 | 2848807 | 284.9672 | 2849092 | 2849422 | 2300753 |
| mod.intI | 640735.1 | 5251.247 | 645986.3 | 657421.2 | 941846.9 |
| mod.intII | 40046.5 | 14499.43 | 54545.93 | 91864.38 | 70889.18 |
| mod.intIII | 41886.02 | 16217.6 | 58103.63 | 97675.84 | 76402.72 |

**Table 3. Posterior mean and 95% Credibility interval for fixed effect of $\alpha$.**

| Year | Parameter | Estimate | SD | LL | UL |
|---|---|---|---|---|---|
| 2012 | $\sigma_u^2$ | 0.2703 | 0.0877 | 0.1414 | 0.4822 |
| | $\sigma_v^2$ | 0.2407 | 0.0278 | 0.19 | 0.2991 |
| $Frac_{spatial}$ | $Var_u/(Var_u + \sigma_v^2)$ | 52% | | | |
| 2013 | $\sigma_u^2$ | 0.2696 | 0.0822 | 0.1454 | 0.4657 |
| | $\sigma_v^2$ | 0.2668 | 0.0309 | 0.2107 | 0.332 |
| $Frac_{spatial}$ | $Var_u/(Var_u + \sigma_v^2)$ | 49% | | | |
| 2014 | $\sigma_u^2$ | 0.2942 | 0.0889 | 0.159 | 0.5059 |
| | $\sigma_v^2$ | 0.2775 | 0.0307 | 0.2216 | 0.3421 |
| $Frac_{spatial}$ | $Var_u/(Var_u + \sigma_v^2)$ | 50.5% | | | |
| 2015 | $\sigma_u^2$ | 0.3981 | 0.1455 | 0.1925 | 0.7558 |
| | $\sigma_v^2$ | 0.2928 | 0.0318 | 0.2342 | 0.3594 |
| $Frac_{spatial}$ | $Var_u/(Var_u + \sigma_v^2)$ | 56% | | | |
| 2016 | $\sigma_u^2$ | 0.6749 | 0.2602 | 0.3131 | 1.3203 |
| | $\sigma_v^2$ | 0.3877 | 0.0392 | 0.3153 | 0.4691 |
| $Frac_{spatial}$ | $Var_u/(Var_u + \sigma_v^2)$ | 62% | | | |
| 2017 | $\sigma_u^2$ | 0.4947 | 0.1602 | 0.2576 | 0.8805 |
| | $\sigma_v^2$ | 0.4135 | 0.0442 | 0.3326 | 0.5059 |
| $Frac_{spatial}$ | $Var_u/(Var_u + \sigma_v^2)$ | 53% | | | |
| 2018 | $\sigma_u^2$ | 0.3466 | 0.1016 | 0.192 | 0.5879 |
| | $\sigma_v^2$ | 0.4846 | 0.0615 | 0.3735 | 0.6147 |
| $Frac_{spatial}$ | $Var_u/(Var_u + \sigma_v^2)$ | 41% | | | |
| 2012-2018 | $Frac_{spatial}$ | | | | |
| | $Var_u/(Var_u + \sigma_v^2)$ | 52% | | | |

SD:Standard Deviation, LL: Lower Level, UL: Upper Level

greater than 15. Those four sectors were in the City of Kigali, with the highest RR observed in Gasabo District Gikomero sector (RR = 19.6, 95% CI = 19.13, 20.05). The two other sectors were in the Southern province Kigoma sector in Nyanza District (RR = 19.7, 95% CI = 19.23, 20.25) and Gikonko sector in Gisagara District with a RR 16.8, 95% CI = 16.42, 17.20). In the Eastern province in, Nyagatare District, was Nyagatare sector with a RR = of 15.75, (95% CI = 15.51, 16.01). This indicates that the malaria cases are concentrated in a few areas, while the disease rate is low in most sectors.

In 2013, there was an increase in the number of sectors with RR ranging between one and four, an increase of 22.12% as compared to 2012. In 2014, 36 (8.65%) sectors had RR > 4. For the year 2015, 39.9% sectors had a RR > 1, and 5.3% of sectors had a RR > 4. In 2016, 40.87% of all the sectors had RR >1 and 6.97% of sectors had RR > 4. In 2017, 37.98% of sectors had a RR > 1 and 9.13 of sectors had a RR > 4. Similar to the previous year, in 2018 37.74% of sectors had RR > 1 and 7.69% of sectors had RR > 4. In conclusion, compared to the overall risk in the year 2012, the risk has increased in later years. In addition, the number of sectors with lower than average risk in the year 2012 decreased over time.

### 3.3 Assessment of Malaria policy to reduce incidence in Rwanda

Rwanda Malaria's strategic plan 2012-2018 [5] aimed to reduce malaria incidence by 20% in 2015 and 42% in 2018. These results show the probability taking into account spatial

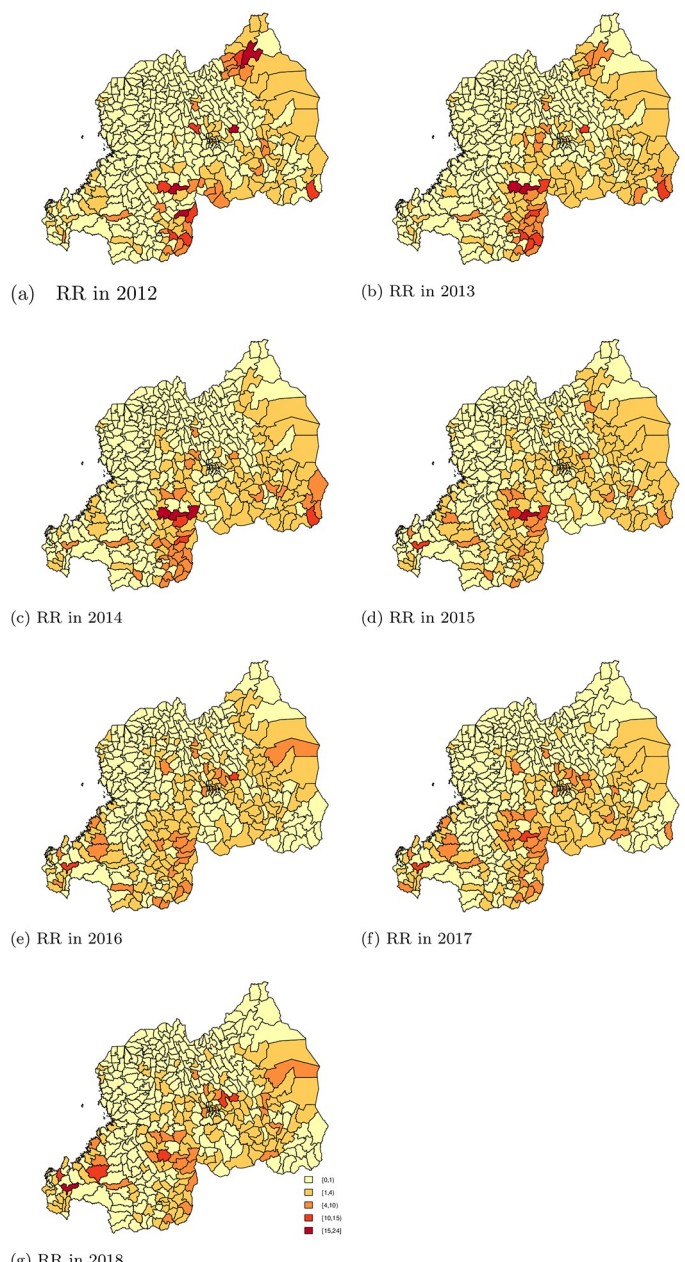

(a)   RR in 2012

(b) RR in 2013

(c) RR in 2014

(d) RR in 2015

(e) RR in 2016

(f) RR in 2017

(g) RR in 2018

**Fig 2. Malaria relative risk from year 2012 to 2018.**

uncertainty as it provides local details of the spatial variation of the risk. Figs 4 and 5 present the area-specific probabilities of failing to reach the target goals. Areas colored red have a high probability (above 80%) having failed to reach the target goal, while areas in yellow have high probability (above 80%) of having successfully reached the target goal. For areas in orange, we are uncertain about whether or not the sectors succeeded in achieving the target goals.

At the baseline year 2012, 29.33% (122) and 33.65% (140) of sectors had a high probability ($> 0.80$) of having a smaller than average risk ($< 0.58$ and $< 0.80$, respectively). The number of sectors that failed to reach the target of 20% reduction increased over the years. Similarly, the number of sectors that failed to reach the target of 42% also increased over the years.

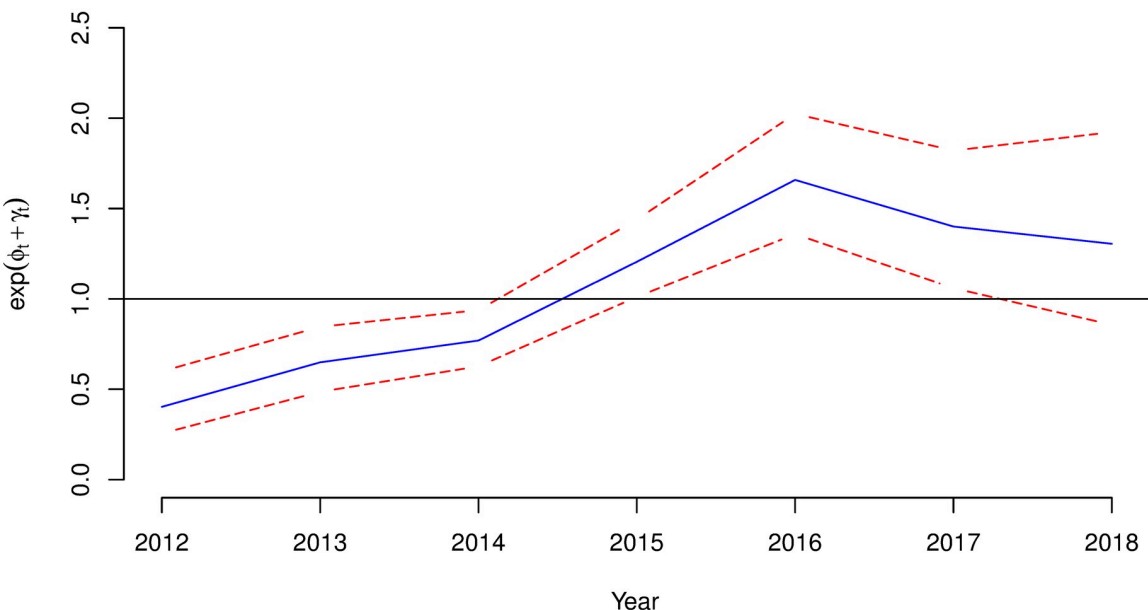

**Fig 3. Posterior temporal trend effect for malaria relative risk in Rwanda: Exp($\phi_t + \gamma_t$) with 95% credible interval.**

**Table 4. Malaria RR per year as compared to the year 2012.**

| Year | [0,1) | [1,4) | [4,10) | [10,15) | [15,24) |
|---|---|---|---|---|---|
| 2012 | 307(73.80%) | 75(18.03%) | 23(5.53%) | 7(1.68%) | 4(0.96%) |
| 2013 | 290(69.71%) | 92(22.12%) | 25(6.01%) | 7(1.68%) | 2(0.48%) |
| 2014 | 278(66.83%) | 102(24.52%) | 30(7.21%) | 3(0.72%) | 3(0.72%) |
| 2015 | 250(60.10%) | 144(34.62%) | 18(4.33%) | 3(0.72%) | 1(0.24%) |
| 2016 | 246(59.13%) | 141(33.89%) | 27(6.49%) | 2(0.48%) | 0(0%) |
| 2017 | 258(62.02%) | 120(28.85%) | 36 (8.65%) | 2(0.48%) | 0(0%) |
| 2018 | 259(62.26%) | 125(30.05%) | 26(6.25%) | 5(1.20%) | 1(0.24%) |

This is due to increased malaria incidence across all the sectors from 2012 to 2016. In 2017 and 2018, the incidence decreased, but did not reach levels lower than the incidence 2012. While there was an improvement in progress towards reaching the target in some years for certain areas, the improvement did not persist over the entire follow-up period. After insecticide residual spry (IRS) intervention in 2015, 2016 and, 2017 the sectors of Nyagatare (North-East) and Kirehe (South-East) showed a reduction in incidence. At the same time, we see that in the South-West, while targets were reached in the earlier years, these areas failed to sustain progress. Table 5 shows a summary of the number of sectors that did not achieve the targets set out by Rwanda's malaria strategic plan with a certain probability.

## 4 Discussion

Spatial data has increased substantially due to the advances in computational tools that allow collection and integration of diverse real-time data sources. This goes in hand with the development of less or complex innovative statistical models to deal with the spatial structure of data in hand [21]. Model-based statistical methods are useful in low resource settings for

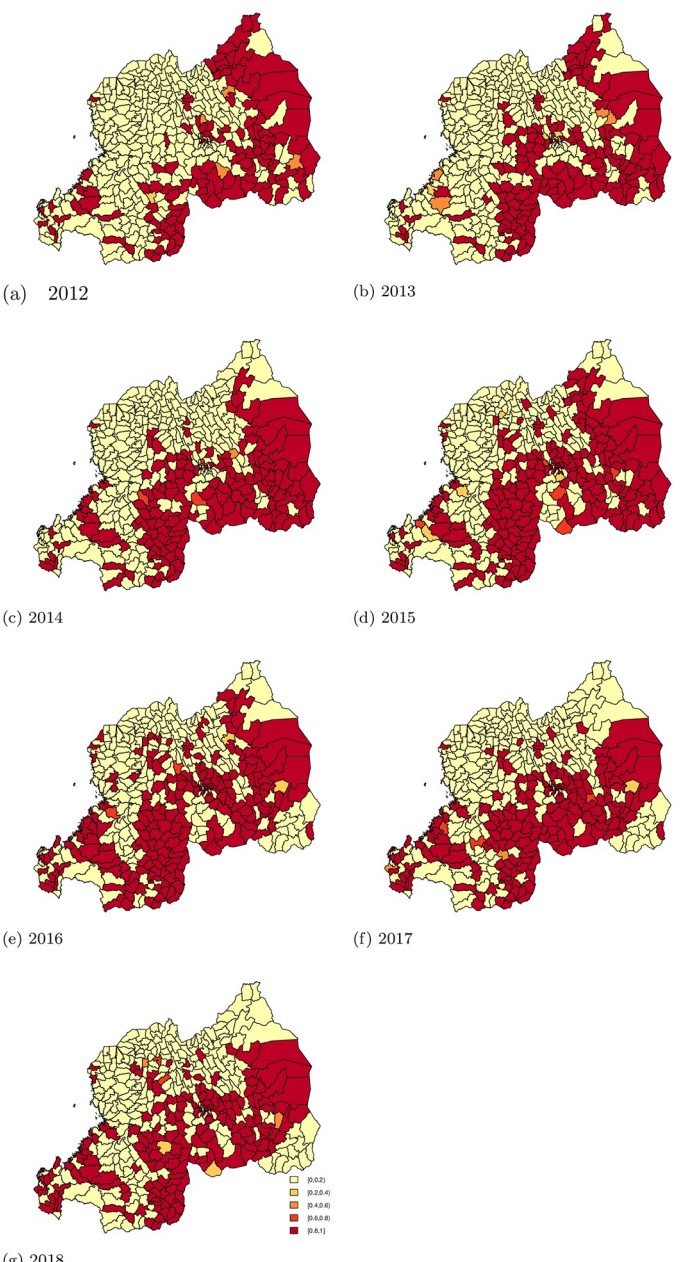

**Fig 4. The area-specific probabilities of not reaching the target goal of 2015 (reduction of 20% as compared to 2012).**

estimating disease risk by health decision-making units as well as in the analysis of uncertainty for survey data [30]. In this paper, the utility of model based statistical methods in estimating the probability of reaching specific targets is presented.

In the past, data quality concerns restricted the use of health facility data as a source of population based statistics. Introduction of web-based information systems for health facility data and the implementation of universal health policy contributed the completeness and accuracy of data at local level and population-based statistics based on those data. This success was prompted by the intensive monitoring of sustainable development goals [31, 32]. Data from

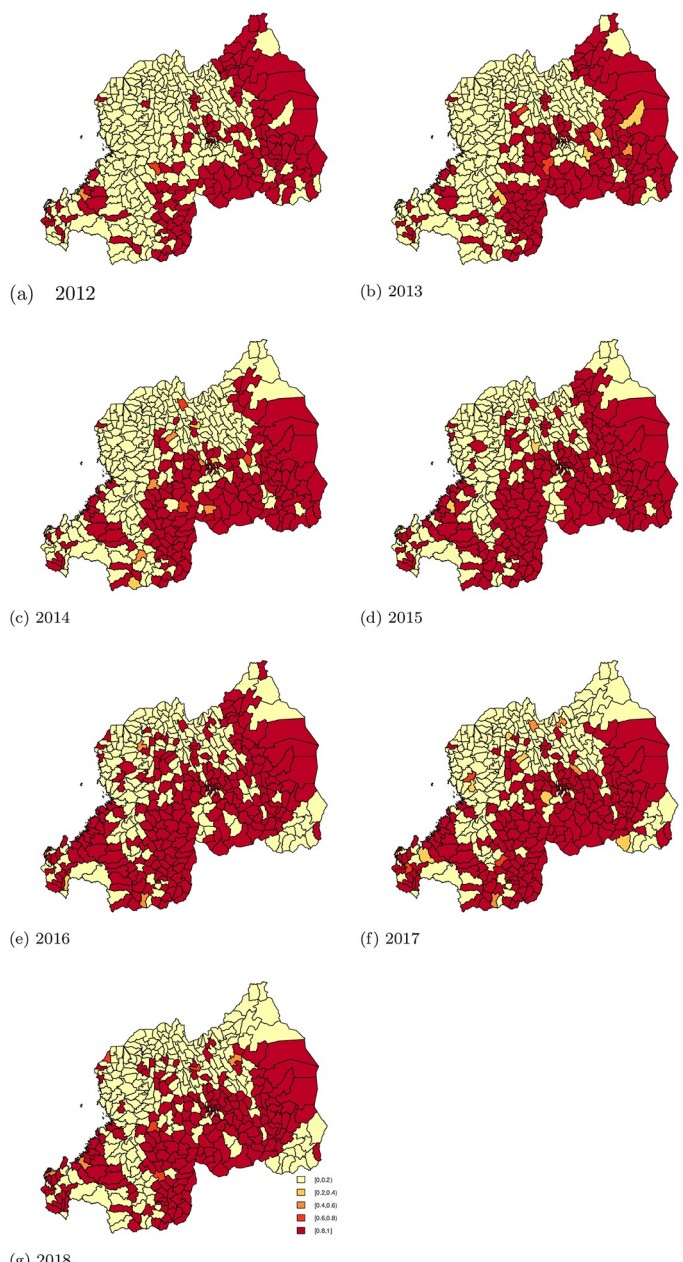

**Fig 5. The area-specific probability of not reaching the target goal of 2018 (reduction of 42% as compared to 2012).**

health facilities in Rwanda are generally of high quality, though successfully integrating these data into health policy and decision-making throughout the health system is an ongoing challenge. [33].

The spatial modeling analysis for malaria data in Rwanda suggested an overall increase in relative risk (RR) in almost all sectors of Rwanda from 2012 to 2016, with a slight decrease from 2017 and 2018. The number of sectors with RR > 1 increased tremendously. In some sectors, the RR was above 10. This implies that malaria incidence increased considerably over time in all sectors of Rwanda but the increase was not consistent over the years.

**Table 5. The sectors that did not achieve reducing the targets.**

| Year | Target of reducing 20% | | | | |
|------|----------|----------|----------|----------|----------|
|  | [0,0.2) | [0.2,0.4) | [0.4,0.6) | [0.6,0.8) | [0.8,1) |
| 2012 | 289(69.47%) | 1(0.24%) | 4(0.96%) | 0(0%) | 122(29.33%) |
| 2013 | 271(65.14%) | 4(0.96%) | 4(0.96%) | 0(0%) | 137(32.93%) |
| 2014 | 258(62.02%) | 1(0.24%) | 1(0.24%) | 4(0.96%) | 152(36.54%) |
| 2015 | 222(53.37%) | 4 (0.96%) | 0(0%) | 5(1.20%) | 185(44.47%) |
| 2016 | 218(52.40%) | 3(0.72%) | 0(0%) | 2(0.48%) | 193(46.39%) |
| 2017 | 236(56.73%) | 2 (0.48%) | 2(0.48%) | 3(0.72%) | 175(41.59%) |
| 2018 | 241(57.93%) | 2(0.48%) | 2(0.48%) | 2(0.48%) | 169(40.62%) |
|  | Target of reducing 42% by 2018 | | | | |
| 2012 | 273(65.62%) | 1(0.24%) | 1(0.24%) | 1(0.24%) | 140(33.65%) |
| 2013 | 250(60.10%) | 3 (0.72%) | 2(0.48%) | 2 (0.48%) | 159(38.22%) |
| 2014 | 235(56.49%) | 3(0.72%) | 6(1.44%) | 4(0.96%) | 168(40.38%) |
| 2015 | 200(48.08%) | 2 (0.48%) | 0(0%) | 0(0%) | 214(51.44%) |
| 2016 | 187(44.59%) | 1(0.24%) | 2(0.48%) | 0(0%) | 226(54.33%) |
| 2017 | 203(48.80%) | 6(1.44%) | 4(0.96%) | 3(0.72%) | 200(48.08%) |
| 2018 | 216(51.92%) | 0(0%) | 3(0.72%) | 4(0.96%) | 193(46.39%) |

The estimated probability of achieving the target for reduction of malaria incidence showed that, almost half (47.36%) of all sectors failed to meet the target of reducing 42% of malaria incidence by 2018, with 80% or 90% certainty. Contrary to expectations from the Malaria Strategic plan [5], malaria incidence increased in East, South, Central, and South-West of Rwanda. Those areas of Rwanda are known as high malaria risk zones [5]. This means that the malaria control program should concentrate efforts on reducing transmission through preventive interventions such as indoor residual spraying (IRS) and bed-net distribution. As Figs 4 and 5 demonstrate in 2013, 2015, 2016 and, 2017 in the North East (Nyagatare) and South East (Kirehe); the reduction may be due to the IRS intervention that occurred in the same period in those Districts. With 90% probability, 51.92% of sectors reduced malaria incidence as planned; however, those sectors belong in Northern provinces and North-West of Rwanda where malaria cases are often lower than other parts of Rwanda. Despite this encouraging success, much work remains to reduce the incidence of malaria across the country. Implementing pre-elimination strategies in those sectors should be premature, instead the focus should be implementing malaria control strategies.

The results presented here are based of malaria cases from health facilities and the population distribution, and the database had limited variables that could have been included in the analysis to explain increased relative risk and the reasons for failing to achieve the target of reducing incidence as planned. We limited our scope on statistical method to evaluate reduction of malaria incidence using an excess probability approach. This approach is a relevant tool to guide decision makers and develop health policy. This model and results can contribute to improvement of malaria surveillance to ensure implementation of interventions in the right place and at the right time.

A disease like malaria requires a strong surveillance system that can enable a quick response to any changes in behaviors related to malaria. Efficient algorithms that can be deployed in response to real-time data collection and make inferences would contribute to a fast response to potential public health threats. [15]

## 5 Conclusion

In summary, we recommend the approach of using spatio-temporal models and routinely collected facility-based data to assess achievement of targets related to malaria incidence and estimate malaria relative risk at the local level. This approach enables us to generate maps that provide information about the probability and uncertainty of reaching the targets, as well as providing information on the spatial contribution to malaria burden in the country. The proposed approach is not only limited to malaria data, but can also be applied in other areas of health care delivery. Spatio-temporal specifications with interactions of both time and space were considered but were not successful.

This era of sustainable development goals (SDGs), especially SDG 3 and its target 3.3 of ending malaria by 2030, requires a tool like the one presented here for planning, monitoring, and evaluation. The excess probability can be applied to survey or routine data from health facilities. It uses routine data efficiently to permanently monitor the changes in malaria transmission and evaluate progress towards national targets. Though survey data are important, provided that data quality are high, routinely collected data are collected more frequently and thus provide more timely assessments of health burden. Many surveys only publish new evidence every five years (such as Demographic and Health survey) and often do not provide estimates at a local level.

## Supporting information

**S1 Data.**
(R)

**S2 Data.**
(R)

**S3 Data.**
(R)

## Acknowledgments

The authors are grateful to Rwanda Biomedical Center, Malaria and other Parasitic Diseases division, for collaboration and discussion on Rwanda's Malaria's strategic plan evaluation approaches. We are grateful for Rwanda malaria resource materials shared by Dr. Aimable Mbituyumuremyi, Malaria Division manager and technical support provided by Mr. Hamza Ndabateze, HMIS officer, to extract data from Rwanda health information system (HMIS).

## Author Contributions

**Conceptualization:** Muhammed Semakula, Christel Faes.

**Data curation:** Muhammed Semakula.

**Formal analysis:** Muhammed Semakula, Christel Faes.

**Funding acquisition:** Muhammed Semakula.

**Investigation:** Muhammed Semakula.

**Methodology:** Muhammed Semakula, Christel Faes.

**Project administration:** Muhammed Semakula.

**Resources:** Muhammed Semakula, Christel Faes.

**Software:** Muhammed Semakula, Christel Faes.

**Supervision:** François Niragire, Christel Faes.

**Validation:** Muhammed Semakula, François Niragire, Christel Faes.

**Visualization:** Muhammed Semakula, Christel Faes.

**Writing – original draft:** Muhammed Semakula.

**Writing – review & editing:** Muhammed Semakula, François Niragire, Christel Faes.

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
