## [Decision Letter · Decision Letter 0]

26 Nov 2019

PONE-D-19-30728

Bayesian spatial modeling of malaria risk in Rwanda

PLOS ONE

Dear Mr Semakula,

Thank you for submitting your manuscript to PLOS ONE. After careful consideration, we feel that it has merit but does not fully meet PLOS ONE’s publication criteria as it currently stands. Therefore, we invite you to submit a revised version of the manuscript that addresses the points raised during the review process.

I would like to point out that both reviewers have stated that the methodology used in the paper is only partly sound and I concur with them. The major issue that the authors should address in their revision is the improvement of the spatio-temporal analysis, which is currently carried out by fitting a model year by year. This is not just unconventional in spatio-temporal modelling, it is also less statistically efficient than developing a spatio-temporal model that exploits correlation between years. Unless there are scientific reasons for this, which are not explained, I cannot see any justification for the adopted modelling approach.  This point will be crucial in deciding whether the re-submitted manuscript will be rejected or not.

We would appreciate receiving your revised manuscript by Jan 10 2020 11:59PM. To enhance the reproducibility of your results, we recommend that if applicable you deposit your laboratory protocols in protocols.io, where a protocol can be assigned its own identifier (DOI) such that it can be cited independently in the future. For instructions see: http://journals.plos.org/plosone/s/submission-guidelines#loc-laboratory-protocols

We look forward to receiving your revised manuscript.

Kind regards,

Emanuele Giorgi

Academic Editor

PLOS ONE

Journal Requirements:

1. We note that you have stated that you will provide repository information for your data at acceptance. Should your manuscript be accepted for publication, we will hold it until you provide the relevant accession numbers or DOIs necessary to access your data. If you wish to make changes to your Data Availability statement, please describe these changes in your cover letter and we will update your Data Availability statement to reflect the information you provide.

Reviewers' comments:

Reviewer's Responses to Questions

**Comments to the Author**

1. Is the manuscript technically sound, and do the data support the conclusions?

Reviewer #1: Partly

Reviewer #2: Partly

2. Has the statistical analysis been performed appropriately and rigorously? 

Reviewer #1: No

Reviewer #2: No

3. Have the authors made all data underlying the findings in their manuscript fully available?

Reviewer #1: No

Reviewer #2: No

4. Is the manuscript presented in an intelligible fashion and written in standard English?

Reviewer #1: Yes

Reviewer #2: No

5. Review Comments to the Author

Reviewer #1: Routine data are becoming increasingly important as countries adopt DHIS2 across sub-Sahara Africa. The approaches presented in the paper are important to allow national malaria control programmes assess progress (or lack of it) at sub national units of decision making. I have the following comments.

1. What’s the fidelity of the data used: were there facilities that were no reporting in some months or not reporting at all? The authors should summarize the completeness of data (how many facilities reported 12 times in year? How many did not report at all?

2. Clarify on the manuscript if all data used was lab confirmed

3. Why was the sector used as the unit of analysis? is this the unit of decision used by the national malaria control programme?

4. Often individuals are assigned based on the health facility they attend as opposed to their residential locations in DHIS2. This means the risk in certain sector is not the actual risk because it’s based on patients from that sector and the neighbouring sectors. Can the authors elaborate in the Rwanda case and how they handle this issue?

5. How were neighbouring areas defined? (queen, rook, based on distance etc.?) and why the choice? It has been shown different choices yield different results

6. Likewise, why the BYM model? they are several other choices (Besag, Besag2, BYM2, Leroux CAR, proper CAR etc.) BYM2 handles identifiability and scaling better

7. The choice of priors and justification is not provided

8. Why were different models fitted by year instead of a spatio-temporal model to also account for temporal correlation and where necessary space-time interactions?

9. The first two paragraphs of 3.2 should be in the methods section

10. Why were there no covariates used to assist in the estimation process?

11. Are the data used (as indicated) to model and accompanied code provided? Coukld not find the URL to either of the two

Reviewer #2: Review comments

The paper addresses an important issue on using population data and clinical routine malaria data for decision making in control/elimination of malaria. The paper uses a model based approach in the analysis and mapping of malaria incidence in Rwanda, for the period from 2012 to 2018.

A) Minor comments

a. Abstract: specify the SDG number that is being referred to there.

b. Abstract: “The results showed an increase of risk of malaria and 47.36% of sectors in Rwanda” is not very clear. Is this increase national or in specific sectors only?

c. Line 9:10, do the authors mean “categorised” when they mention “situated”?

d. Line 14: Most countries “placed”….

e. Line 36 – 37: …finally increasing %... use “percentage” instead of the symbol.

f. Line 45: For the “epidemiological”….

g. Line 46 – 47: Re-write to make it clear.

h. Line 54 – 55: …. Patterns in low areas transmission should change to …. “patterns in low transmission areas”.

i. Line 91 – 92: Can authors specify what percentage of the total was not included in the final analysis?

j. Line 97 – 102: Authors need to indicate by properly subscripting the count for time as rightly done for geographic areas in the same paragraph.

k. In Table 1, can the authors add footnotes (or in the caption) to explain what SD, LL and UL are?

l. Line 228: Change “sound statistics” to “accurate statistics”.

m. Line 234 – 236 is not clear in the current form.

n. Line 237: This in the current form is misleading. Should it be reading: .... almost half (47.36%) of all sectors....

o. On lines 246 – 249, what threshold is being looked at here? It’s not coming out very clearly.

p. Figures are not (properly) captioned, making it difficult to follow or align the text to the Figures.

i. The Figure (Fig1_Desc2), titled “The under-five Malaria 2012 0 2018” with no sex specified seems redundant as it is adding very little information. The information presented in the graph can be explained in the text.

ii. The Y-axis on Fig1_Desc5 “Overall malaria per year” is misleading. The Figure needs to be re-done to properly convey the correct information.

iii. The legends in Figures 2 – 4 should be properly positioned, interfering with maps currently.

q. Go through the manuscript to correct typos and grammatical errors.

B) Major comments

a. The language in the current form of the paper still needs extensive editing to make things clear.

i. Abbreviations are not properly defined throughout the document i.e. figure 3 instead of Figure 3 etc.

ii. Capitalisation is not properly used throughout the document.

iii. Several sentences are not very clear as indicated in the minor comments above.

iv. Several places missing commas and full stops – distorting the message

v. Inappropriate tenses used.

vi. Inappropriate use of directions i.e. “east north” as indicated on lines 237 - 250 instead of “North east”

b. Lines 119 – 122: Authors indicate that they use Bayesian methods for the analysis, and. Have taken the time to explain both the data and process models. However, conspicuously missing are details on the priors used in model fitting.

c. Line 132 – 136: Authors introduce the concept of calculating policy relevant threshold. Three issues arise here.

i. How is the threshold “c” determined or reached at? This is not clearly explained in the document. For the reader to understand the policy relevant goals, determination of these thresholds needs to be clearly explained.

ii. In line 135, authors claim that: “If |P| is large, the set goal is likely reached in that area.” What is |P| being compared to?

iii. In P(Theta_it < (100 - c)%), is the relative risk based on the observed data? If so, this has to be explicitly stated.

d. The authors have 7 years worthy of data. Enough data to enable a proper spatio-temporal model. However, in lines 157 – 159, they indicate that they have fitted separate models for each year. I find this problematic in the sense that they are underutilising the data and not fully leveraging the information therein. A spatio-temporal model will enable borrowing strengths in the data across both space and time, therefore giving a complete picture of how malaria incidence has changed since the base year of 2012. The model fitted currently has huge implications on the conclusions authors draw in the paper. I comment on this in later sections below.

e. In Tables 2 and 4 and lines 174 – 192, authors present malaria relative risk per year as compared to base year of 2012. They group RR into 5 groups: 0-1, 1 – 4, 4 – 10, 10 – 15, and 15 – 24. They use square braces. This presents several challenges in the sense that:

i. The mathematical/statistical meaning of these braces means that these groups are not distinct, meanwhile in text, these groups of RR are presented as distinct. Authors should not [0, 1] and [0, 1) will mean two different things.

ii. Again, for example, column one has [0, 1] and column two has [1, 4]. Does this mean that these two RR groups contain both the 1? This is same for all the groups and it is misleading.

iii. More confusing is the fact that in the text, they resort to using round braces. For reasons in point 1 above, this becomes more confusing.

iv. Therefore lines 184 – 192 need to be re-written with correct presentation of Table 2.

f. In section 3.3, lines 193 - 215 authors present an “assessment of Malaria policy to reduce incidence in Rwanda.” With this goal of analysis in mind, it makes more sense to use spatio-temporal model, so that the available data take into account the trends leading up to the target year (2015 and 2018) for the target non-exceedance thresholds of 20% and 42% respectively. See comment in point (d) above.

g. Lines 203 – 205: The authors should endeavor to quantify this increase, for it to be helpful and relevant to policy makers.

h. Lines 206 – 209 should be re-written to properly convey the message contained in there. More importantly, the claims raised in these lines can be affirmed by using a proper spatio-temporal analysis in relation to the concerns raised in points (d) and (f) above.

i. Line 237 – 239. Authors claim that almost half (47.36%) of the sectors did not meet the targets with 80 or 90% certainty. What would be helpful is for the authors to show clearly each of these certainties on map. See, for example:

i. Giorgi et. al. (2018), Using non-exceedance probabilities of—relevant malaria prevalence thresholds to identify areas of low transmission in Somalia. Malar J. 2018;17:88.

ii. Macharia et. al. (2019), Spatio-temporal analysis of Plasmodium falciparum prevalence to understand the past and chart the future of malaria control in Kenya

iii. Yankson et. al. (2019), Geostatistical analysis and mapping of malaria risk in children under 5 using point-referenced prevalence data in Ghana

j. In their discussion, on lines 250 – 251, authors mention that “Implementing pre-elimination strategies in those sectors should be considered consciously.” With the the incidence presented here, it’s not proper for the authors to start talking about pre-elimination. The message to policy implementers should rather be to focus on control strategies at this point.

k. On lines 258 – 259, authors mention that “It can contribute to improve Malaria surveillance to ensure appropriate intervention in the right place and most needed time.” This is very correct, but based on the statistical analysis presented here, authors should be cautious in their statements on conclusions made. A proper spatio-temporal analysis would be required to make this conclusion on time.

6. PLOS authors have the option to publish the peer review history of their article (what does this mean?). If published, this will include your full peer review and any attached files.

Reviewer #1: Yes: Peter M Macharia: KEMRI Wellcome Trust Research Programme

Reviewer #2: No

---

## [Author Response · Author response to Decision Letter 0]

29 Jan 2020

Response to the reviewers are attached in separate letter

---

## [Decision Letter · Decision Letter 1]

18 Feb 2020

PONE-D-19-30728R1

Bayesian spatio-temporal modeling of malaria risk in Rwanda

PLOS ONE

Dear Mr Semakula,

Thank you for submitting your manuscript to PLOS ONE. After careful consideration, we feel that it has merit but does not fully meet PLOS ONE’s publication criteria as it currently stands. Therefore, we invite you to submit a revised version of the manuscript that addresses the points raised during the review process.

The revised submission is  a substantial improvement over the the original submission and the authors are on the right track. I see two outstanding issues which I believe the authors can successfully address.

 1) Provide a model selection approach that is not not exclusively based on indices as those used in Table 2. One approach might be to identify models that give more precise and accurate predictions as well as reliable 95% confidence intervals using a hold out sample. I am not convinced that  model.intIV has extremely large values for the the chosen indices due to numerical instability.

2) Provide a more clear explanation of the spatio-temporal models. For example the verb "to combine" is too vague and it is not clear what that mean mathematically. Also, the fact that some covariance matrices have Kroncker products is because the authors have multiplied main spatial and temporal effects (in which case "to combine" means "to multiply")? The authors could provide more explanation of this either in the main manuscript or in the supplementary material, as they prefer. 

In the revision, please also consider any of the remaining points raised by one of the reviewers. 

We would appreciate receiving your revised manuscript by Apr 03 2020 11:59PM. To enhance the reproducibility of your results, we recommend that if applicable you deposit your laboratory protocols in protocols.io, where a protocol can be assigned its own identifier (DOI) such that it can be cited independently in the future. For instructions see: http://journals.plos.org/plosone/s/submission-guidelines#loc-laboratory-protocols

We look forward to receiving your revised manuscript.

Kind regards,

Emanuele Giorgi

Academic Editor

PLOS ONE

Reviewers' comments:

Reviewer's Responses to Questions

**Comments to the Author**

1. If the authors have adequately addressed your comments raised in a previous round of review and you feel that this manuscript is now acceptable for publication, you may indicate that here to bypass the “Comments to the Author” section, enter your conflict of interest statement in the “Confidential to Editor” section, and submit your "Accept" recommendation.

Reviewer #1: All comments have been addressed

Reviewer #2: (No Response)

2. Is the manuscript technically sound, and do the data support the conclusions?

Reviewer #1: Yes

Reviewer #2: Partly

3. Has the statistical analysis been performed appropriately and rigorously? 

Reviewer #1: Yes

Reviewer #2: Yes

4. Have the authors made all data underlying the findings in their manuscript fully available?

Reviewer #1: No

Reviewer #2: Yes

5. Is the manuscript presented in an intelligible fashion and written in standard English?

Reviewer #1: Yes

Reviewer #2: No

6. Review Comments to the Author

Reviewer #1: My comments have been addressed satisfactorily and the paper is now in a format that can be published

Reviewer #2: I appreciate the opportunity to review this revised paper. The authors have resolved some of the initial problems in the analysis, particularly the issue of using an appropriate modelling approach for the kind of data and objectives of analysis at hand. There are however still a few issues remaining. See attached

7. PLOS authors have the option to publish the peer review history of their article (what does this mean?). If published, this will include your full peer review and any attached files.

Reviewer #1: Yes: Peter Macharia: KEMRI Wellcome Trust Research Programme

Reviewer #2: No

---

## [Author Response · Author response to Decision Letter 1]

21 Apr 2020

Dear Reviewer,

Thank you for your comments. We have addressed all the comments and response to the reviewer letter is attached.

---

## [Editor Report · Decision Letter 2]

28 Apr 2020

PONE-D-19-30728R2

Bayesian spatio-temporal modeling of malaria risk in Rwanda

PLOS ONE

Dear Mr Semakula,

Thank you for submitting your manuscript to PLOS ONE. After careful consideration, we feel that it has merit but does not fully meet PLOS ONE’s publication criteria as it currently stands. Therefore, we invite you to submit a revised version of the manuscript that addresses the points raised during the review process.

The revised submission is  a substantial improvement over the the original submission and the authors are on the right track. I see two outstanding issues which I believe the authors can successfully address.

 1) Provide a model selection approach that is not not exclusively based on indices as those used in Table 2. One approach might be to identify models that give more precise and accurate predictions as well as reliable 95% confidence intervals using a hold out sample. I am not convinced that  model.intIV has extremely large values for the the chosen indices due to numerical instability.

2) Provide a more clear explanation of the spatio-temporal models. For example the verb "to combine" is too vague and it is not clear what that mean mathematically. Also, the fact that some covariance matrices have Kroncker products is because the authors have multiplied main spatial and temporal effects (in which case "to combine" means "to multiply")? The authors could provide more explanation of this either in the main manuscript or in the supplementary material, as they prefer.

We would appreciate receiving your revised manuscript by Jun 12 2020 11:59PM. To enhance the reproducibility of your results, we recommend that if applicable you deposit your laboratory protocols in protocols.io, where a protocol can be assigned its own identifier (DOI) such that it can be cited independently in the future. For instructions see: http://journals.plos.org/plosone/s/submission-guidelines#loc-laboratory-protocols

We look forward to receiving your revised manuscript.

Kind regards,

Emanuele Giorgi

Academic Editor

PLOS ONE

Additional Editor Comments (if provided):

I did provide comments in the previous decision letter from the journal but I notice these have not been considered. I paste those comments below.

I see two outstanding issues which I believe the authors can successfully address.

1) Provide a model selection approach that is not not exclusively based on indices as those used in Table 2. One approach might be to identify models that give more precise and accurate predictions as well as reliable 95% confidence intervals using a hold out sample. I am not convinced that model.intIV has extremely large values for the the chosen indices due to numerical instability.

2) Provide a more clear explanation of the spatio-temporal models. For example the verb "to combine" is too vague and it is not clear what that mean mathematically. Also, the fact that some covariance matrices have Kroncker products is because the authors have multiplied main spatial and temporal effects (in which case "to combine" means "to multiply")? The authors could provide more explanation of this either in the main manuscript or in the supplementary material, as they prefer.

---

## [Author Response · Author response to Decision Letter 2]

15 Jun 2020

Response to the reviewer’s comments

Manuscript Title: Spatio-temporal modeling for malaria risk in Rwanda

Date: May 30, 2020

1. Provide a model selection approach that is not not exclusively based on indices as those used in Table 2. One approach might be to identify models that give more precise and accurate predictions as well as reliable 95% confidence intervals using a hold out sample. I am not convinced that model.intIV has extremely large values for the the chosen indices due to numerical instability..

Response: 

Thank you for your advice. Indeed, you are right, the model.intIV did not converge. It was not chosen neither discussed in this manuscript due to convergence issues. We decided to remove the model.intIV from the Table 2 since it is not part of final findings and DIC are not reasonable.

In addition to DIC, sensitivity analysis for model selection, we have added Conditional predictive ordinate, that split data into the two groups. The model is run on yf so that posterior distribution for the parameters p(θ| yf ) is obtained; R-INLA runs the so-called leave one out cross validation ,which assumes that yf =y_i yc =yi. The CPO was computed in R-INLA. 

2. Provide a more clear explanation of the spatio-temporal models. For example the verb "to combine" is too vague and it is not clear what that mean mathematically. Also, the fact that some covariance matrices have Kroncker products is because the authors have multiplied main spatial and temporal effects (in which case "to combine" means "to multiply")? The authors could provide more explanation of this either in the main manuscript or in the supplementary material, as they prefer.

Response:

This comment is relevant. More clarification was provided on sptio-temporal models, and particularly on Kronecker products. The explanation is provided on lines 140-149.

In fact, there are Kronecker products because there is interaction between space and time. Therefore, the structure matrix is factorized as a Kronecker product of corresponding main effects which interact. The detailed materials are provided in R codes annexed.

Thank you very much for all your comments. We are very grateful.

---

## [Editor Report · Decision Letter 3]

18 Jun 2020

PONE-D-19-30728R3

Bayesian spatio-temporal modeling of malaria risk in Rwanda

PLOS ONE

Dear Dr. Semakula,

Thank you for submitting your manuscript to PLOS ONE. After careful consideration, we feel that it has merit but does not fully meet PLOS ONE’s publication criteria as it currently stands. Therefore, we invite you to submit a revised version of the manuscript that addresses the points raised during the review process.

We look forward to receiving your revised manuscript.

Kind regards,

Emanuele Giorgi

Academic Editor

PLOS ONE

Additional Editor Comments (if provided):

I think this is an improvement over the previous version but there are still some aspects that, in my view, still require a major revision.

- More explanation about the condition predictive ordinate needs to be provided. Simply mentioning that this available in INLA is not enough and does not provide the reader with enough information to assess the validity of this approach.

- The explanation given for the models in Table 1 is not clear. Especially, it is impossible for the reader to understand what is the difference between the models from Type II to Type IV.

- Discarding model Type IV because of failure to converge is not a valid justification because it also implies that using a different fitting algorithm may indeed lead to convergence. However, a plausible explanation may also be that the model of Type IV is indeed too complex for the data. The authors should think of how to show evidence that this is indeed the case. In a non-Bayesian context, a natural approach would be to show that the profile likelihood for the covariance parameters is flat and the 95% confidence interval based on that is indeed too wide.

---

## [Author Response · Author response to Decision Letter 3]

30 Jul 2020

Response to Reviewer’s comments

1. - More explanation about the condition predictive ordinate needs to be provided. Simply mentioning that this available in INLA is not enough and does not provide the reader with enough information to assess the validity of this approach.

Answer: 

Thank you for this comment, we have provided more explanation from line 144 to 155 to enable the reader to have enough information and understanding on CPO

2. -The explanation given for the models in Table 1 is not clear. Especially, it is impossible for the reader to understand what is the difference between the models from Type II to Type IV.

Answer: 

Thank you, we have provided more explanation on the table 1 by adding three paragraphs. Line 156 to 183. We explained each type of interaction in details to enable the reader to have a good understanding of each type of model interaction (Type I to IV)

3. - Discarding model Type IV because of failure to converge is not a valid justification because it also implies that using a different fitting algorithm may indeed lead to convergence. However, a plausible explanation may also be that the model of Type IV is indeed too complex for the data. The authors should think of how to show evidence that this is indeed the case. In a non-Bayesian context, a natural approach would be to show that the profile likelihood for the covariance parameters is flat and the 95% confidence interval based on that is indeed too wide.

Answer: 

Thank you, though model Type IV converged the variance is too small (Stdev: 0.00269982) due to 

overspecification. Therefore, it is hard to see the differences as compared to type II.

---

## [Editor Report · Decision Letter 4]

31 Jul 2020

PONE-D-19-30728R4

Bayesian spatio-temporal modeling of malaria risk in Rwanda

PLOS ONE

Dear Dr. Semakula,

Thank you for submitting your manuscript to PLOS ONE. After careful consideration, we feel that it has merit but does not fully meet PLOS ONE’s publication criteria as it currently stands. Therefore, we invite you to submit a revised version of the manuscript that addresses the points raised during the review process.

We look forward to receiving your revised manuscript.

Kind regards,

Emanuele Giorgi

Academic Editor

PLOS ONE

Additional Editor Comments (if provided):

The last remaining point is that the Type IV specification of the sptatio-temporal correlation.

Fail to convergence in the INLA software does not invalidate a model.

The authors should consider two options in their revision: 1) explaining this issue more in details and provide evidence of overspefication, which is completely absent from the paper; 2) remove this model from the methods and explaining in the conclusion that spatio-temporal specifications with interactions were considered but were not successful.

In both option, please, provide a clear definition of "overspefication" without assuming the reader is familiar with this concept.

---

## [Author Response · Author response to Decision Letter 4]

6 Aug 2020

Response to Reviewer’s comments

The last remaining point is that the Type IV specification of the sptatio-temporal correlation. Fail to convergence in the INLA software does not invalidate a model.

The authors should consider two options in their revision: 1) explaining this issue more in details and provide evidence of overspefication, which is completely absent from the paper; 2) remove this model from the methods and explaining in the conclusion that spatio-temporal specifications with interactions were considered but were not successful.

In both option, please, provide a clear definition of "overspefication" without assuming the reader is familiar with this concept.

Answer:

Thank you for this comment, as you suggested we have removed TYP IV model from the methods and we provided explanations in conclusion as suggested. 

( lines 171- 175, 223-223 in track change version were removed ) and (lines 337 and 338 were added in manuscript)

The term overspecification does not appear anywhere in paper more since it was related the type IV model.

---

## [Editor Report · Decision Letter 5]

19 Aug 2020

Bayesian spatio-temporal modeling of malaria risk in Rwanda

PONE-D-19-30728R5

Dear Dr. Semakula,

We’re pleased to inform you that your manuscript has been judged scientifically suitable for publication and will be formally accepted for publication once it meets all outstanding technical requirements.

Kind regards,

Emanuele Giorgi

Academic Editor

PLOS ONE
---

## [Editor Report · Acceptance letter]

24 Aug 2020

PONE-D-19-30728R5 

Bayesian spatio-temporal modeling of malaria risk in Rwanda 

Dear Dr. Semakula:

I'm pleased to inform you that your manuscript has been deemed suitable for publication in PLOS ONE. Congratulations! Your manuscript is now with our production department. 

Kind regards, 

on behalf of

Dr. Emanuele Giorgi 

Academic Editor

PLOS ONE